# Long-Term Measurement of Piglet Activity Using Passive Infrared Detectors

**DOI:** 10.3390/ani11061607

**Published:** 2021-05-29

**Authors:** Roberto Besteiro, Tamara Arango, Juan Ortega, María D. Fernández, Manuel R. Rodríguez

**Affiliations:** 1BeWell Agrosolutions, 27004 Lugo, Spain; tamara.arango@bewellagrosolutions.com; 2Xunta de Galicia, Oficina Agraria Comarcal Lalín, Rúa Areal, 27, 36500 Pontevedra, Spain; juan.antonio.ortega.martinez@xunta.es; 3Department of Agroforestry Engineering, University of Santiago de Compostela, Escola Politécnica Superior de Enxeñaría, Campus Terra, s/n, 27002 Lugo, Spain; mdolores.fernandez@usc.es (M.D.F.); manuelramiro.rodriguez@usc.es (M.R.R.)

**Keywords:** animal activity, passive infrared detector, piglets, wavelet analysis, daily pattern

## Abstract

**Simple Summary:**

The activity of weaned piglets on a commercial farm was measured using passive infrared detectors during six breeding cycles. Highly heterogeneous daily behaviors were observed. A high level of movement was detected immediately after weaning, followed by a sudden drop and a subsequent stabilization throughout the cycle. The real series were best characterized by models with a single peak of activity.

**Abstract:**

Measuring animal activity is useful for monitoring animal welfare in real time. In this regard, passive infrared detectors have been used in recent years to quantify piglet activity because of their robustness and ease of use. This study was conducted on a commercial farm in Northwest Spain during six complete breeding cycles. The hourly average activity of weaned piglets with a body mass of 6–20 kg was recorded and further analyzed by using a multiplicative decomposition of the series followed by a wavelet analysis. Finally, the real series were compared to the theoretical models of activity. Results showed a high level of movement immediately after weaning and a sustained level of activity throughout the cycles. The daily behavior of the piglets followed a clear circadian pattern with several peaks of activity. No differences in behavior were observed between spring–summer cycles and autumn–winter cycles. Single-peak models achieved the best predictive results. In addition, the installed sensors were found to underestimate mild activity.

## 1. Introduction

Social changes in recent years, brought about by climate change, access to continuous and updated information or the growing intensification of the livestock industry, have led to a change in the behavior of consumers of agrifood products, who are now more concerned about the environment and animal welfare. In the near future, these changes will require a continuous assessment of each housing environment and of the animals living in it according to animal-welfare-relevant indicators and a precise documentation of results. Yet, already established criteria for assessing animal welfare are often resource- and management-based [1].

Actually, animal welfare indicators can be divided into environment-based indicators (housing, feeding, room temperature and humidity, etc.) and animal-based indicators (possible injuries, body condition, behavioral change, etc.). In recent years, animal-based indicators have gained increasing attention and are now considered by researchers to be essential for a comprehensive evaluation of welfare in its multidimensionality [2,3]. Thus, innovative animal-based sensor systems for continuous and objective monitoring and evaluation of the behavior of groups of animals like pigs could make use of, for example, visually acquired information by using appropriate sensors in combination with learning algorithms or similar applications [1].

In this context, factors such as animal activity and behavior emerge as useful indicators for establishing a level of animal welfare on livestock farms and become particularly relevant in intensive production systems, which restrict some behaviors considered as natural by ethology [4]. For example, pen areas with higher levels of ammonia concentration will tend to be less occupied than other areas [5]. Therefore, animal activity measurements in these areas will be lower than in cleaner areas. Similarly, high temperatures will compel animals to be less active and even to rest in fouling areas [6,7,8].

Various methods have been used for measuring animal activity. Most authors have used camera technology (49%), followed by microphones (18%), animal-attached sensors (15%), including accelerometers and RFID (radio frequency identification) tags or other sensors (16%), such as passive IR detectors (PID) [9]. The general suitability of the PID method for monitoring the activity of weaner pigs [4,10,11,12,13] and fattening pigs [14] has been tested in various studies. The PID method, assessed against human observation of a group of weaned piglets, has shown more limited in terms of behavior detection but cheaper for farmers (equipment and mounting cost less than 100 €). Accordingly, PID could be more widely implemented [15].

In the above studies, activity data referred to large groups of animals or were undifferentiated for all animals in a compartment and less focused on smaller groups of pigs in a pen or in special areas within the pen [1]. Likewise, behavioral observation methods can be a source of important information on certain focus areas, insofar as these methods provide patterns of animal activity, which has been used as an indicator of animal disease. For instance, a change in the drinking behavior [16] or increased activity on the enrichment material could be indicative of the beginning of tail biting [17]. Thus, PIDs could possibly focus on specific areas of the pen to monitor behavioral changes [1].

So far, studies using the PID method have focused on short periods, but have not tested the suitability of PIDs during long periods of continuous measurements. Actually, current theoretical models have been designed for short periods of time. Consequently, the long-term functionality of the method has not yet been demonstrated.

The objective of this paper is to characterize the activity of weaned piglets during transition from 6 to 20 kg live weight and to define the daily cyclic components of activity for six production cycles. To this end, activity data were obtained by using the method described in [4], which consisted in using commercial PIDs without altering their output signal. In order to explain the daily behavior of the animals and determine the exogenous variables that affect animal behavior under real farm conditions, the derived time series was analyzed by spectral analysis using the fast Fourier transform (FFT) and wavelet analysis.

## 2. Materials and Methods

### 2.1. Animals and Housing

To obtain animal activity data, an experimental test was conducted on a commercial pig farm in Northwest Spain (ED50: 43°10′15” N, 8°19′24” W). The farm housed weaned piglets of 6 kg to 20 kg live weight and had a maximum capacity of 4985 sows. The weaner room (Figure 1), with an area of 69.26 m^2^ and a volume of 164.50 m^3^, consisted of twelve 2.55 m × 1.97 m pens on both sides of a central aisle. The room could hold a maximum of 300 piglets, with an area of 0.20 m^2^ per piglet. In this case, the piglets were commercial hybrids obtained by mating a Landrance-Large White F1 sow with a German Piétrain boar, weaned at 21 days. The piglets grew from 6 to 20 kg live weight during a postweaning transition of approximately 41 days.

The floor was completely slatted over a pit with 45 cm of depth. The ventilation system consisted of a 500 mm helical extractor fan with a volume of 8746 m^3^ h^−1^. Fan speed was adjusted by changing the voltage using a temperature-based digital controller, which allowed ventilation rates of between 25% and 100% and bandwidth temperatures of ±1.5 °C.

Additionally, the ventilation rate was modulated with a manual system that reduced the area of the air outlet through the fan and provided volumes of between 1.03 and 10.58 m^3^ h^−1^ per piglet. Fresh air entered the room through two 0.70 m^2^ windows with manually controlled air deflectors. The radiant floor heating system comprised two 1.205 m × 0.50 m polyester spreader plates for water, with a capacity of 19 L, placed at the centre of each pen. The average temperature of the plates was 30.60 °C, with a mean difference of 5.80 °C between inlet and outlet temperature. The heating system was controlled with a manual valve.

Animal activity (Aa) was monitored during six production cycles between October 2011 and May 2013. The cycles had an average duration of 41 days each and were separated by a period for sanitary emptying. Animal activity is defined here as every movement made by the animals while performing active behaviors such as walking, feeding or interacting with pen mates. Animal activity was measured using OPTEX RX-40QZ passive infrared detectors placed over the entrance door at a height of 2.30 m, with 12.00 × 12.00 m^2^ coverage, ±0.020% sensitivity and a normally closed (NC) relay output (Figure 1). The output was activated for 2.5 s when the sensor detected activity such that the alarm condition was reached, i.e., when the sensor computed 4 triggers in 20 s. Consequently, the proposed method quantified animal activity by computing the time during which the alarm was activated for a certain period. Times of activity were totaled and stored by a data logger at 10-min intervals (Campbell Scientific Ltd. CR-10X). The data logger recorded the PIR detector alarm time with 0.125 s accuracy [4].

### 2.2. Data Analysis

The activity time series obtained from the PID was analyzed using the R Team (2015) integrated development environment and the R Core Team (2016) “stats” package. First, hourly average activity data were obtained from the 10-min intervals. Then, the series was decomposed into the trend, seasonal and random components using moving averages and a multiplicative model. The trend component was determined using a moving average and was then subtracted from the series. The seasonal component was computed by averaging, for each time of the day, over all days. Finally, the random component was the remainder term of the original time series. Daily repetitive behaviors were determined by removing the random component of the series.

A wavelet analysis was performed to understand the dynamics of the series of animal activity in the time and frequency spectrum. This analysis broke up the signal into shifted and scaled versions of the original wavelet function, called a mother wavelet. To conduct the analysis, we used the “WaveletComp” package [18], and chose the Morlet Wavelet as the mother wavelet because the wavelet scale was almost equal to the Fourier period [19]. Accordingly, a continuous Morlet wavelet transform was performed. A frequency sampling resolution of 1/20 suboctave per octave was established.

In this study, the models of animal activity described in [4] were used because the measurement method was the same, except that the sensor covered the entire room area from a height of 2.30 m, whereas in [4] the sensor covered a single pen and was placed at a height of 0.80 m.

Additionally, to compare the data series for animal activity derived from the theoretical models, a standardization of the series was performed by subtracting the average and dividing by the standard deviation. The standardized value was compared to the patterns of activity defined in [4] for Equations (1) and (2) peaks of activity:Aa1(t) = 1 + 0.54cos(2π/24t + 2.41)(1)
Aa2(t) = 1 + 0.49cos(2π/24t + 2.46) + 0.32cos(4π/24t + 2.06) + 0.25cos(6π/24t − 1.35)(2)

Predictions were evaluated using the following metrics: root mean square error (RMSE), mean absolute relative error (MARE) and correlation coefficient “r” between each of the cycles and models. Additionally, a Student’s *t*-test for repeated measures was performed upon having verified normality through the Shapiro test.

## 3. Results and Discussion

A preliminary visual inspection of the time series plot showed that animal activity followed a circadian cycle in every one of the six time series, with an increase in movement during daylight hours and rest during night periods (Figure 2). Actually, this was the expected behavior, as it is widely described in the literature [13,20,21].

In addition, visual inspection revealed that the levels of activity recorded in this study never reached the theoretical maximum of 600 s, with a peak value of 336 s (56% of the maximum, Table 1). The basal level of activity was near 0 s for every cycle, especially during the night period, with significant peaks of activity during daylight hours (Figure 2). Moreover, PID sensors tend to slightly overestimate animal activity [1].

This behavior contrasted with the behavior described in [4], which showed more stability, with less significant peaks and less evident basal activity. Such a difference can be attributed mainly to the differences in the measurement methods used. The method reported in this study captured more intense group activity events more easily, but showed a greater difficulty in recording the moments with a lower level of activity. This phenomenon could be explained by the loss of sensitivity of the PID sensors with distance [14] and the creation of blind spots of measurements caused by physical obstacles and the measurement angle of the sensor.

### 3.1. Series Decomposition and Analysis of the Trend and Seasonality Components

The six time series for animal activity were decomposed into the trend, seasonal and random components using a multiplicative seasonal model. The analysis of the trend component revealed a high level of movement immediately after weaning, which then sharply decreased (Figure 3). This behavior has also been described in [4,14], and is associated to a stressful event for the animals caused by either the abrupt separation from the mother, the creation of new social groups, aggressions or exploration of new housing. These behaviors gradually decrease after weaning [22], most noticeably during the first 2–3 days.

Afterwards, a long period of stability was observed, particularly during the spring–summer cycles (Figure 3), with a slight upward trend. The analysis of the cycles did not reveal a negative impact of the increase in live mass/m^2^ on animal activity, even though density reached the maximum allowable limit in the European Union, 0.20 m^2^/piglet of up to 20 kg [23]. This finding is in contrast with the results reported by [1], who found a reduction in the levels of activity over time, with animals starting at 23 kg and ending at 85 kg of carcass weight and housed at a density of 1.39 m^2^/pig, thus exceeding the density recommended by the European Union, set at 0.65 m^2^/pig of 85 to 110 kg of live weight. Likewise, Ref. [4] found a decrease in the levels of activity of piglets of less than 20 kg. Probably, the age of the animals affects their predisposition to exploratory behavior. Therefore, it would be interesting to identify the changes in animal behavior and determine in which moment of their life cycle such changes occur. Yet, in the present study, an increase in animal activity was observed at the end of cycle C4. There is no obvious explanation for this event, although it should be noted that 45 animals (15%) were removed from the room for commercial purposes 27 days after weaning during cycle C4. During the rest of the cycles, only a few animals were removed due to health problems. Possibly, both the handling operations and the increase in the free area available per animal were responsible for the increase in animal activity.

In terms of seasonality, no differences were found in the levels of daily activity between spring–summer cycles and autumn–winter cycles. In contrast, Ref. [24] found higher activity values in cold weather than in warm weather in a swine finishing house. Undoubtedly, temperature plays a key role in animal activity, and animals become more lethargic in warmer environments. Moreira et al. [25] observed the effect of temperature on the suckling behavior of piglets and found that the duration of the suckling behavior was shorter in summer than in winter, and the piglets suckled more often during the night. This strategy and others such as adapting their lying or eating behaviors are used by pigs to counterbalance the effects of being out of their thermoneutral zone. Le Dividich and Herpin [26] observed that the increase in heat production in cold is associated with an increase in the turnover of both FFA (free fatty acid) and glucose that involves changes in thyroid hormones and catecholamines. Finally, Ref. [27] evaluated whether different early life thermal stressors (ELTS) altered the temperature preference of pigs later in life and the future thermoregulation.

In this study, the animals were housed in an environmentally controlled facility seasonal. Accordingly, variations in temperature are likely to be lower.

The analysis of the seasonal component (Figure 4) revealed a low level of activity during the hours of darkness. This phenomenon was expected because of the circadian behavior of the piglets [20], with much less activity during the night [20] as compared to the activity recorded during daylight hours, thus following a cycle of 24-h periodicity. Moreover, as stated above, the measurement method used neglected those events with less activity, which predictably occur more often during the night.

All the studied cycles showed peaks of activity during daylight hours, which occur mainly during the morning in the spring–summer cycles and during the evening in the winter cycles (Figure 4). Nevertheless, defining a standard activity curve for every cycle is not possible because the peaks of activity or rest throughout the day are heavily influenced by the management routines established on each farm, among which the time assigned to feeding or inspecting the animals. In this respect, Refs. [28,29] reported up to 70% of feed intake during daylight hours, and 30% at night.

In recent years, many authors have defined daily patterns of activity in pigs with either a single wide peak of activity or two peaks, mainly distributed in the morning and evening [4,24,30,31]. Although these models are widespread, these patterns do not always exactly match the hourly data recorded throughout our six study cycles. Similar results were reported by [14] for the first month after weaning, during which PID sensors recorded multiple peaks of activity. Similarly, Ref. [1] defined a pattern with multiple peaks of activity during the first 19 days of a cycle of pig fattening, which then evolved towards a pattern with two peaks of activity from day 20 of the cycle.

### 3.2. Wavelet Analysis

In order to obtain an overview of the evolution of the main frequencies of the harmonics included in the series over time, a wavelet analysis was performed on every series.

As shown in Figure 5, a periodicity of 24 h was observed in every series throughout every cycle, particularly C3 and C4, in which the maximum power values were observed along the 24 h-periodicity line. This finding supports the results for the seasonal component in relation to the circadian rhythm of the animals. After the 24 h periodicity, the 12 h periodicity was the first harmonic of the fundamental wave of the 24 h period. The 12 h periodicity occasionally gained relevance throughout the cycles, especially on C3 and C5. Adding this first harmonic generated a daily pattern of activity that changed from a wide peak of activity to two peaks of activity with a period of less activity between both peaks (Figure 5). Finally, a periodicity of 8 h, which corresponded to the second harmonic, occurred intermittently and with little relevance throughout the cycles, as in C1 and C4 (Figure 5). Adding this second harmonic accentuated the two peaks of activity in the behavioral pattern. For values over 24 h, the wavelet analysis did not reveal periodicities that accounted for the composition of our data series. Below the fundamental wave, a greater level of noise was found, with multiple periodicities occurring in every cycle over time.

The importance of these three periodicities was already suggested by [4], although the weight of the two harmonics of the fundamental wave throughout the cycle was greater in that case. Furthermore, Ref. [4] observed a clear difference between the first half and the second half of the cycle with regard to the behavioral pattern. Contrastingly, such a difference was found in any of the six cycles analyzed in the present study.

Based on the different seasonal components extracted from the time series for animal activity in Section 3.1 and the relevance of the 24 h periodicity revealed by wavelet analysis, the activity of piglets of 6–20 kg was most reliably defined by a sinusoidal wave. Hourly measurements revealed daily patterns of activity with one, two, three or more peaks. Yet, such variations cannot be easily accounted for.

### 3.3. Comparison with Theoretical Models

As shown in Figure 6, the peaks of activity in the standardized series were more marked than the peaks in the theoretical models, which could not provide a full estimation. The lack of sensitivity of the PIDs used in this configuration of the experimental design was the reason behind the high maximum levels of activity found, even in the standardized series. Figure 6 visually demonstrates that the single-peak behavioral pattern provided a slightly better estimation than the two-peak pattern because of the daily variability in the actual measured behavior.

The findings in Figure 6 are supported by the metrics shown on Table 2. The model with a single peak of activity presented a lower RMSE and a higher correlation coefficient. In contrast, no significant differences were found for MARE, in spite of the fact that the sinusoidal model showed a higher value. Yet, these results must be interpreted with caution, given the limited number of cycles.

In this respect, the theoretical model of activity that best characterized the daily variability of piglet behavior was a single-peak model, such as the dromedary model [32]. Nevertheless, Ref. [32] presented the “camel model” with two peaks of activity as a more sophisticated approach. Based on our analysis, more complex patterns must be used when there is evidence that this type of behavior is actually occurring, as reported in [12].

## 4. Conclusions

Based on animal activity data measured with passive infrared detectors (PIDs) in piglets from 6 to 20 kg during six breeding cycles with an average duration of 41 days, group activity was characterized by a marked peak of activity during the first hours after weaning, which was stabilized in a few hours and remained constant for the rest of the cycle. Daily behavior was highly heterogeneous, but it could not be directly assimilated to a single-peak model or a two-peak model, as is commonly suggested in the literature. Nevertheless, for the large dataset used in this study, single-peak models best characterized the six breeding cycles. In addition, using only one sensor for the entire breeding room led to an underestimation of mild activity and an overestimation of active behavior for all the animals.

Using PID sensors is increasingly interesting because of their ease of use, their robustness and the flexibility of measurements, which are reliable and can be easily interpreted. The present study reveals the benefits of validating technologies in real farming conditions for long periods in order to gain deeper knowledge of the behavior of farm animals.

## Figures and Tables

**Figure 1 animals-11-01607-f001:**
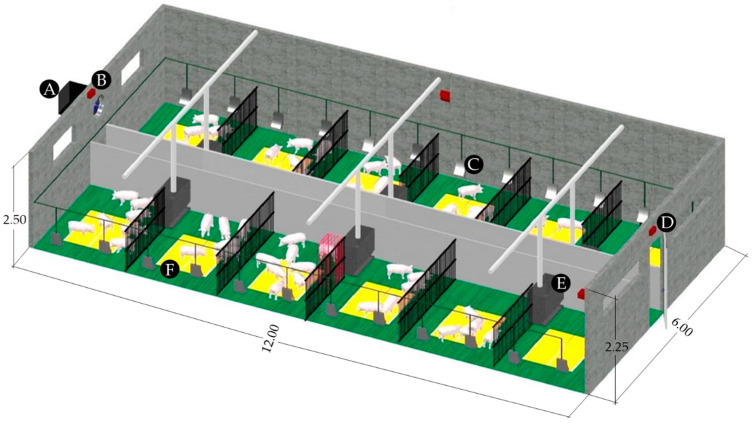
Experimental room and measurement points: **A** = extractor fan, **B** = volume of air extracted, **C** = drinker, **D** = OPTEX RX-40QZ animal activity sensor, **E** = feeder and **F** = heating plate.

**Figure 2 animals-11-01607-f002:**
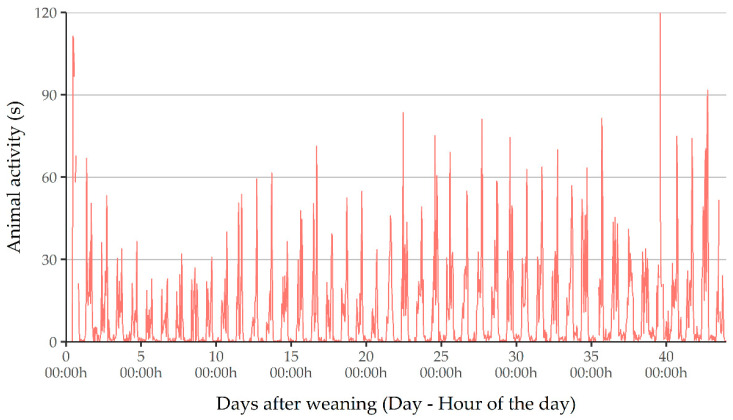
Evolution of animal activity in C2 measured in 10-min intervals.

**Figure 3 animals-11-01607-f003:**
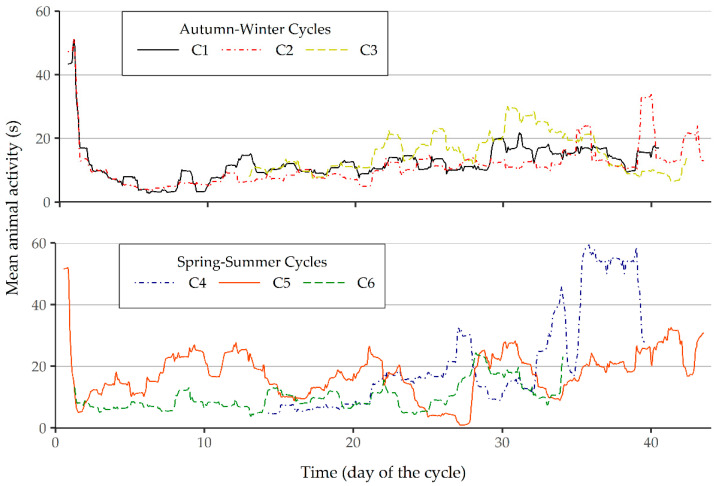
Seasonal component of the multiplicative decomposition of the series of animal activity for the autumn–winter and spring–summer cycles.

**Figure 4 animals-11-01607-f004:**
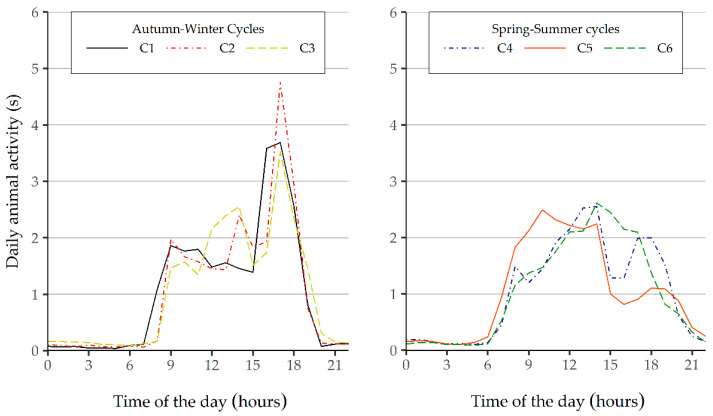
Seasonal component of the multiplicative decomposition of the series of animal activity for autumn–winter and spring–summer cycles.

**Figure 5 animals-11-01607-f005:**
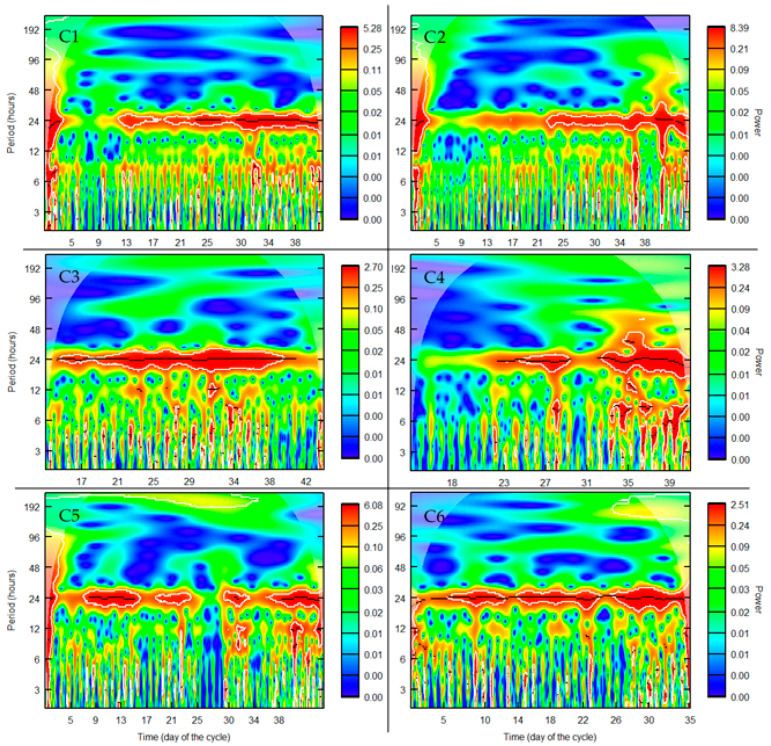
Wavelet analysis of the six cycles of animal activity recorded.

**Figure 6 animals-11-01607-f006:**
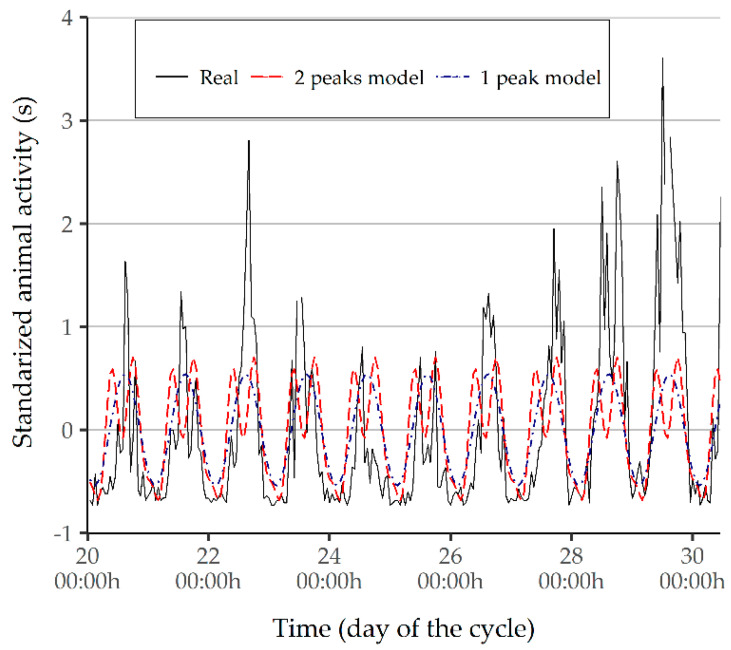
Comparison of the standardized hourly activity between the 1-peak model and the 2-peak model of C6.

**Table 1 animals-11-01607-t001:** Overview of the recorded activity cycles.

Cycle	Mean (s)	Max (s)	Min (s)	SD (s)	Date	Seasons *	Days in Cycle	N Data
C1	12.43	233.50	0.00	21.31	06/10–16/11	A–W	1–42	985
C2	11.60	215.27	0.00	22.04	21/11–04/01	A–W	1–45	1057
C3	15.33	120.80	0.00	19.97	21/01–21/02	A–W	13–44	739
C4	21.35	200.45	0.00	31.82	25/04–22/05	S–S	14–41	631
C5	18.09	336.23	0.00	25.01	31/05–14/07	S–S	1–45	1057
C6	10.36	114.09	0.00	14.22	11/04–15/05	S–S	1–35	812

* Seasons of the cycles: autumn–winter (A–W) and spring–summer (S–S).

**Table 2 animals-11-01607-t002:** Performance of the theoretical models for piglet activity.

	RMSE	r	MARE
1-peak model	0.855	0.543	2.179
2-peak model	0.898	0.435	2.344
*p*-value	<0.01	<0.01	0.304

## Data Availability

The data presented in this study are openly available from FigShare at https://doi.org/10.6084/m9.figshare.14229758.v1 (accessed on 28 May 2021).

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
