# Peer review of "Long-Term Measurement of Piglet Activity Using Passive Infrared Detectors"

_animals, 2021, doi:10.3390/ani11061607_

Round 1

Reviewer 1 Report

The clear relationship between animal activity and animal health  has been reported, and it is necessary to explore a rapid method for the determination of animal activity.  It is recommended to accept after minor revision.

  1. please give a definition of animal activity measured by passive infrared detector? what kind of animal behaviors are included in activity?
  2. Is the unit of axis X wrong in fig 2?
  3. the abbreviation of Autumn-Winter is wrong in title of table 1.
  4. the serial numbers of fig 5 and 6 are wrong
  5. there is no reasonable explanation for the abnormal increase of activity in C4.

Reviewer 2 Report

This is an interesting study, but there appears to be several flaws in the design of this study and the data collection protocols.  Much emphasis is given on the method of data analysis collected using passive infrared sensors, but the results are not correlated to animal's activity and it does not fully represent the actual animal’s behavioral or activity related functions?  

How is animal activity measurement correlated to variations in the ventilation systems or the air quality? This statement is not very clear, and no evidence of correlation is provided or cited from previous work.

Validation of the 'excitement' state with biochemical hormonal analysis or blood-based cortisol or oxytocin or a relevant biomarker test could have strengthened the claims. Variation in the movement activity does not necessarily mean 'excitement'.  Sentence structuring and language needs attention and thorough 'developmental editing' by a native speaker. It is very hard to make a coherent meaning out of the sentences as written in the manuscript draft.

Line 58: (cheaper for farm) - inexpensive? any numbers?

In Figure 1:  What exactly is an animal activity sensor? Is this an RFID tag or the passive infrared detector? (OPTEX RX-40QZ passive infrared 107 detectors)

How did you differentiate the measurements between one pig vs group of pigs using this so called 'animal activity sensor'? The data cannot distinguish whether the measured information is from one pig or herd of pigs? How was this cross validated? 

Round 2

Reviewer 2 Report

Language editing is still needed.  There are several sentences that still does not provide congruent and clear meaning.